# Identifying signal and noise structure in neural population activity with Gaussian process factor models

**Stephen L. Keeley**
Princeton Neuroscience Institute
Princeton University,
Princeton, NJ 08544
StephenLKeeley@gmail.com

**Mikio C. Aoi**
Princeton Neuroscience Institute
Princeton University,
Princeton, NJ 08544

**Yiyi Yu**
Dept. of Electrical
Computer Engineering
University of California Santa Barbara
Santa Barbara, CA

**Spencer L. Smith**
Dept. of Electrical
Computer Engineering
University of California Santa Barbara
Santa Barbara, CA

**Jonathan W. Pillow**
Princeton Neuroscience Institute
Princeton University,
Princeton, NJ 08544

## Abstract

Neural datasets often contain measurements of neural activity across multiple trials of a repeated stimulus or behavior. An important problem in the analysis of such datasets is to characterize systematic aspects of neural activity that carry information about the repeated stimulus or behavior of interest, which can be considered "signal", and to separate them from the trial-to-trial fluctuations in activity that are not time-locked to the stimulus, which for purposes of such analyses can be considered "noise". Gaussian Process factor models provide a powerful tool for identifying shared structure in high-dimensional neural data. However, they have not yet been adapted to the problem of characterizing signal and noise in multi-trial datasets. Here we address this shortcoming by proposing "signal-noise" Poisson-spiking Gaussian Process Factor Analysis (SNP-GPFA), a flexible latent variable model that resolves signal and noise latent structure in neural population spiking activity. To learn the parameters of our model, we introduce a Fourier-domain black box variational inference method that quickly identifies smooth latent structure. The resulting model reliably uncovers latent signal and trial-to-trial noise-related fluctuations in large-scale recordings. We use this model to show that in monkey V1, noise fluctuations perturb neural activity within a subspace orthogonal to signal activity, suggesting that trial-by-trial noise does not interfere with signal representations. Finally, we extend the model to capture statistical dependencies across brain regions in multi-region data. We show that in mouse visual cortex, models with shared noise across brain regions out-perform models with independent per-region noise.

# 1 Introduction

Recent advances in electrophysiological and calcium fluorescence imaging technologies have enabled the collection of increasingly high-dimensional neural datasets. Making sense of such datasets will rely on the development of flexible statistical methods for extracting relevant structure. Gaussian process factor models provide one powerful tool for identifying low-dimensional latent structure from high-dimensional neural response data. These models seek to characterize neural time-series data in terms of a small number of smoothly evolving latent variables, and have been successfully used to characterize neural representations in a variety of contexts [1, 2, 3, 4, 5, 6].

Standard Gaussian Process factor analysis (GPFA) uses a Gaussian process prior to impose smoothness on inferred latent variables, but do not explicitly consider stimulus or task conditions. However, neural data often exist in the form of repeated trials, whereby the same condition is presented to an animal multiple times. These repeated presentations give rise to neural activity that varies across trials around some time-varying "signal" component that is typically estimated using the peri-stimulus time histogram (PSTH). Understanding this signal, and its relationship to trial-to-trial variability, is of central importance to the models of coding in the nervous system [7, 8], yet latent factor models have not been developed to explicitly study this question. Here we address this shortcoming by developing an extension to Gaussian process factor analysis with Poisson spiking (P-GPFA) which we call signal and noise P-GPFA (SNP-GPFA). This model incorporates both signal and independent per-trial components that vary across trials. We refer to these latter components as "noise", in the sense that they are not time-locked to the repeated stimulus, though they may well reflect other signals unrelated to the experimental stimulus of interest.

In both P-GPFA and SNP-GPFA models, because the Gaussian process is not a conjugate prior for a Poisson observation model, posterior inference is intractable in closed form. Variational inference methods have become increasingly common for applications of Gaussian processes [9, 10, 5]. They achieve tractability by approximating the posterior distribution $p_\theta(\mathbf{x}|\mathbf{y})$ with a well-behaved variational distribution $q_\phi(\mathbf{x}|\mathbf{y})$ [11]. For P-GPFA and SNP-GPFA, because the calculation of the expectation under $q_\phi(\mathbf{x}|\mathbf{y})$ of the joint distribution $p_\theta(\mathbf{x}, \mathbf{y})$ is also intractable, we use a 'black-box' approach, which works via sampling of the joint distribution [12].

However, black-box variational inference approaches for Gaussian Process Factor models with long time-series can be computationally cumbersome. Therefore, we introduce a variant of black-box variational inference which uses a Fourier-transformed latent representation that factorizes across Fourier modes. This procedure diagonalizes the Gaussian Process (GP) covariance, avoiding a large matrix inversion during inference, thereby providing speed and computational improvements. We demonstrate the inference technique is fast and flexible in a simpler P-GPFA framework, and then use it to learn the SNP-GPFA model quickly and efficiently.

The SNP-GPFA model recovers separate signal and noise subspaces, which allows us to answer a number of scientific questions regarding these facets of neural activity. Here, we address two scientific questions with SNP-GPFA. **1)** We characterize the overlap between signal and noise subspaces in monkey V1 data, and **2)** We characterize the extent to which noise is shared across cortical region using multi-region neural recordings from rodent V1 and a higher cortical visual region.

For the first, the alignment of subspaces that reflect different aspects of neural activity has been explored in other contexts [6, 13], as well as the characterization of the subspace of neural noise [14]. Previous work suggests that signal and noise subspaces may be orthogonal [14], and such orthogonal representations may preserve neural information [15]. Our model directly addresses this question. Using SNP-GPFA on primate data we find that there is indeed more noise activity orthogonal to the signal subspace than in the signal subspace, particularly when a visual stimulus is present. This suggests that in monkey V1 trial-by-trial variability does not interfere with stimulus encoding.

To address the second scientific question, we include SNP-GPFA analyses on simultaneously recorded visual regions in rodent cortex to ask if trial-varying activity is shared or independent across cortical regions. We compare performance of SNP-GPFA models that varying in their number of shared and independent noise latents across cortical regions. We find that the model that has shared noise latents performs best on cross-validation measures, suggesting trial-by-trial variability has shared structure across cortical regions in the rodent visual system.

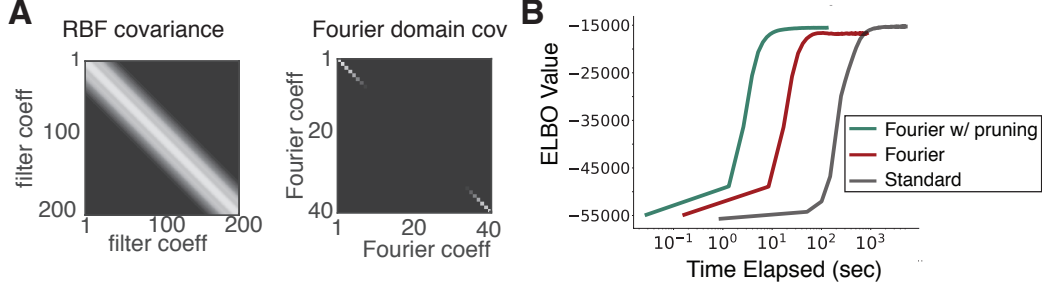

Figure 1: Efficient representation of GP covariance. **(A)** Standard GP covariance matrix for 1D vectorization of 200 timepoints, with length scale $\ell = 15$, and its Fourier representation, pruned. **(B)** Time to maximization of the ELBO in the time-domain inference and in the Fourier domain with and without a minimum frequency.

## 2 Poisson Gaussian Process Factor Analysis (P-GPFA)

We begin by introducing the Poisson-GPFA model, which has been used previously to identify continuous latent states from population spike train recordings [10, 16, 6]. The observations of our model are spike-train data, represented by the neurons-by-time matrix $\mathbf{Y} \in \mathbb{N}^{N \times T}$.

We seek to learn a P-dimensional latent variable $\mathbf{x}(t) \in \mathbb{R}^P$ that linearly maps to the data via a loadings matrix $\mathbf{W} \in \mathbb{R}^{N \times P}$, followed by some nonlinear function $f$ and Poisson observations.

$$\mathbf{Y} = \text{Poiss}(f(\mathbf{W}^\top \mathbf{X})) \tag{1}$$

Our choice of non-linear function $f$ is the softplus $f(x) = \log(1 + \exp(x))$.

Each latent $x_j(t)$ ($j \in \{1 \dots P\}, t \in \{1, 2 \dots T\}$) evolves according to a Gaussian process, $x_j(t) \sim \mathcal{GP}(0, \mathbf{K}(\theta_j))$, with covariance matrix $\mathbf{K}(\theta)$ defined by a squared exponential kernel $[\mathbf{K}(\theta)]_{tt'} = \rho \exp(-|\mathbf{z}(t) - \mathbf{z}(t')|^2/2\ell^2)$, where hyperparameters $\theta = \{\ell, \rho\}$ include a length scale $\ell$ controlling smoothness and a marginal variance $\rho$ controlling magnitude.

Given that the marginal likelihood of this model, $p(\mathbf{Y}|\mathbf{W}) = \int p(\mathbf{Y}|\mathbf{W}, \mathbf{X})p(\mathbf{X}|\theta)d\mathbf{X}$ is not available in closed-form, it is common to use a variational inference approach to learn the parameters of such models [16, 10]. Recall that variational inference seeks to maximize an evidence lower bound (ELBO) using a variational distribution [11]. Here, because the expectation term in the ELBO, $\mathbb{E}_{q_\phi}[\log(p(y|x, w))]$, cannot be calculated analytically, we employ a 'black box approach' which uses Monte-Carlo samples to estimate the expectation term [17]. This inference method is called black-box variational inference (BBVI).

### 2.1 Fourier-domain black-box variational inference

BBVI can be computationally cumbersome. Therefore, to learn the P-GPFA and SNP-GPFA models, we introduce a novel inference method which performs BBVI over a Fourier-represented latent space, which increases both inference tractability and speed. Factorizing and learning the time series in the Fourier domain, rather than the time domain, allows us to take advantage of computational savings conferred by a diagonal covariance matrix while overcoming problems of uncorrelated timepoints which is typical when the variational distribution is factorized over time [18].

Our motivation for this approach is that the GP prior over $x(t)$ describes a stationary process, as its covariance only depends on pairwise distances. This allows us to diagonalize the covariance $\mathbf{K}$ by the Fourier transform (Figure 1A). Here, the covariance matrix $\mathbf{K}$ is diagonalized by $\tilde{\mathbf{K}} = \mathbf{B}\mathbf{K}\mathbf{B}^\top$ where $\mathbf{B}$ is the orthonormal discrete Fourier transform matrix with $[\mathbf{B}]_{\omega,t} = \frac{1}{\sqrt{P}}e^{-\mathrm{i}2\pi\omega t/m}$, $\mathrm{i} \equiv \sqrt{-1}$. The diagonalized kernel is represented as $\tilde{c}(\tilde{\omega}) = \tilde{\rho}e^{-\frac{1}{2}\tilde{\omega}^2\ell^2}$ where $\tilde{\rho} = \sqrt{2\pi}\rho\ell$ is the frequency-domain variance and $\tilde{\omega} = \frac{2\pi}{m}\omega$ represents an adjusted frequency of the GP kernel.

Inference can be conducted completely in the Fourier domain, precluding the need to invert the prior covariance $\mathbf{K}$. The joint likelihood is expressed as

$$\ell(\mathbf{Y}, \mathbf{X}|\mathbf{W}, \theta) = \ell(\mathbf{Y}, \tilde{\mathbf{X}}|\mathbf{W}, \theta) \tag{2}$$

$$\approx \sum_i \mathbf{y}_i \log(f(\mathbf{w}_i^\top \tilde{\mathbf{X}}\mathbf{B})) + \left( f(\mathbf{w}_i^\top \tilde{\mathbf{X}}\mathbf{B}) - \log(\mathbf{y}_i!) \right) \mathbb{1}_T$$

$$- \tfrac{1}{2} \left( Pd \log 2\pi + P \sum_{\tilde{\omega}} \log \tilde{c}_\theta(\tilde{\omega}) + \sum_p \tilde{\mathbf{x}}_p^\top \text{diag}(\tilde{\mathbf{K}}_\theta)^{-1} \tilde{\mathbf{x}}_p \right), \tag{3}$$

where $\tilde{\mathbf{X}}$ represents the Fourier-transformed latents and $\mathbb{1}_T$ is a length-$T$ vector of ones, and $i \in \{1, 2 \dots N\}$ denotes neuron index. The diagonalized representation demonstrably speeds up computational time (Figure 1**B**). Moreover, the inversion of the time-domain $\mathbf{K}$ can present tractability challenges due to computer precision [19], however, the inversion of $\tilde{\mathbf{K}}$ is trivial so long as the vector along the diagonal, $\tilde{\mathbf{K}}$, does not contain values that are too small. When small values are present, we regularize $\tilde{\mathbf{K}}$ by adding a small constant value $(10^{-7})$.

This Fourier-represented GP has additional computational advantages, including methods to prune unnecessary Fourier coefficients that do not substantially contribute to explaining variability in $\mathbf{Y}$. Pruning frequencies constrains the number of coefficients in the Fourier representation to a much smaller number than would be necessary in the time-domain. Pruning of the Fourier representation has the additional consequence of pruning the variational distribution, which shrinks the number of variational parameters. Finally, because Fourier BBVI uses a diagonal Fourier-domain variational distribution, time-correlations are preserved (despite BBVI sampling) due to off-diagonal elements in the time-domain variational distribution.

Fourier methods have been used previously to improve inference for GP models [20, 21, 22], this is to our knowledge the first time this approach has been used with BBVI. Ultimately, this Fourier-domain BBVI method can be viewed as an alternative to many other methods that work to make Gaussian processes computationally efficient, including inducing points and sparse GP approximations [21, 22, 23, 22, 24].

We use our Fourier-domain BBVI to learn the Fourier latents $\tilde{\mathbf{X}}$ via direct optimization of the variational distribution $q_\phi(\tilde{\mathbf{X}})$, factor loading parameters $\mathbf{W}$, and hyperparameters $\ell$. (Note there is an invariance between hyperparameter $\rho$, and the loadings matrix $\mathbf{W}$, so we need not directly learn $\rho$ in this model.) The speed up from optimization with Fourier domain BBVI can be realized most starkly in the domain where the time-series is very long. Figure 1**B** compares Fourier-domain inference to time-domain inference for a Poisson observation GPFA model with a single latent (i.e. $P = 1, T = 1500$) and $N = 10$ neurons. Inference is sped up by conversion to the Fourier domain as the bottleneck in time-domain inference is the inversion of a $1500 \times 1500$ covariance matrix $\mathbf{K}$. By additionally specifying a minimum frequency, sufficiently small frequencies are pruned and the variational distribution and prior covariance can be cut from 1500 values to 62. This provides an additional substantial speed advantage of approximately an order of magnitude. It is important to note that the speed-up of our method depends on the specifics of the number of neurons, latents, latent length, and pruning. For subsequent analyses in this paper, the speed-up of BBVI due to the Fourier-domain implementation is anywhere from 20-70%.

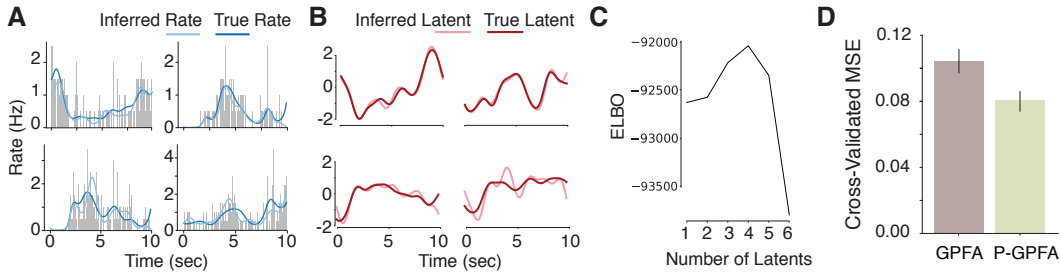

Figure 2: Learning a P-GPFA latent model using Fourier-domain black box variational inference

We validate our Fourier BBVI inference procedure on simulated and real data. Figure 2**A, B** and **C** demonstrate accurate recovery of latent structure and of firing rates on simulated P-GPFA data. Here,

30 neuronal firing rates are generated from a four-dimensional GP-latent space. Figure 2**A** shows the learned and true rates of four simulated neurons. Grey bars indicate spike PSTHs. Figure 2**B** shows the four generative latents and learned latents from Fourier BBVI rotated to their optimal mapping via regression. Figure 2**C** demonstrates that the ELBO value after inference is maximal when using the true number of latents.

The non-conjugacy of P-GPFA (and SNP-GPFA), and thus the reason we need to use the sophisticated inference of Fourier-BBVI, is due to the fact that the observations are Poisson, as opposed to Gaussian. This is an important choice as Poisson observations better describe neural data. We show, using data from rodent visual cortex, the cross-validated mean squared error of the inferred spike rate to smoothed spike rate from held-out trials (Figure 2**D**). The model with Poisson observations performs significantly better that the GPFA model with Gaussian observations. Others have noted similar advantages to Poisson observation factor models for neural data in other settings [10]. For this reason we wish to use a Poisson observation characterization for our SNP-GPFA model.

## 3   SNP-GPFA

To isolate noise and signal subspaces in the P-GPFA framework, we introduce a model that includes separate noise and signal latent structure (SNP-GPFA). We assess the model first on simulated data, and then on two neural data sets. The first of the datasets contains multi-neuron spiking activity from 65 neurons recorded in primate V1 during passive viewing of a drifting sinusoidal grating stimulus, with 72 different orientations for $D = 35$ repeated trials. The second consists of spiking activity from 67 neurons from two regions of rodent visual cortex, recorded during passive viewing of $D = 20$ repeats of a 32-second sinusoidal grating stimulus. Gratings had 8 different orientations which persisted for 4 seconds each. For more information on the data, see [25, 26] and the supplemental materials.

The SNP-GPFA model describes neural activity on trial $j$ as

$$\mathbf{y}_j = \text{Poiss}(f(\mathbf{W_s}^\top \mathbf{X}^s + \mathbf{W_n}^\top \mathbf{X}_j^n)) \tag{4}$$

where $P$ signal latents are drawn from a "signal" Gaussian process, $\mathbf{x}_p^s \sim \mathcal{GP}(0, \mathbf{K}_s)$ with covariance $\mathbf{K}_s$ and concatenated to form $\mathbf{X}^{s\top} = (\mathbf{x}_1^s, \mathbf{x}_2^s, \ldots, \mathbf{x}_P^s)$, which are shared across trials. On each trial, $Q$ independent noise latents are drawn from a "noise" Gaussian process, $\mathbf{x}_q^n \sim \mathcal{GP}(0, \mathbf{K}_n)$ with covariance $\mathbf{K}_n$, forming $\mathbf{X}^{n\top} = (\mathbf{x}_1^n, \mathbf{x}_2^n, \ldots, \mathbf{x}_Q^n)$. Loading weights $\mathbf{W_s}^\top$ and $\mathbf{W_n}^\top$ parametrize a mapping from the dimensionality of the signal space $P$ or the noise space $Q$ to the full $N$-dimensional neural response space. Thus, $\mathbf{W_s}$ is of size $P \times N$ and $\mathbf{W_n}$ is $Q \times N$. Covariance matrices $\mathbf{K}_s$ and $\mathbf{K}_n$ are constructed by evaluating a radial basis covariance at all pairs of time points in a trial. The SNP-GPFA model is outlined schematically in Figure 3**A**. For clarity, we visualize the firing rate of one neuron across trials with only one signal and noise latent dimension.

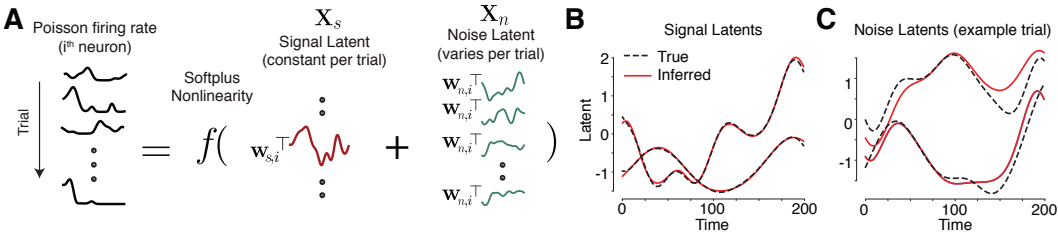

Figure 3: (**A**) Schematic of the SNP-GPFA model. (**B**, **C**) Recovering signal and noise latent structure in simulated SNP-GPFA data.

To perform inference for the SNP-GPFA model, we develop a variational approach similar to that for P-GPFA. We use a variational distribution $q_\phi$ for the latents, parametrized as a fully-independent multivariate normal distribution of dimension $\tilde{T}(P+QD)$ where $\tilde{T}$, which corresponds to the number of Fourier coefficients needed to represent the signal. We determine $\tilde{T}$ by assuming a minimum length scale of 10 ($\ell \geq 10$), which substantially shrinks the number of Fourier coefficients required to represent the latent signal and noise processes (from 321 to 44 dimenions for rodent data, and from

511 to 108 dimensions for primate data). This choice is appropriate, as we typically we do not learn length scales smaller than this value.

To validate the model fit, we show that the Fourier BBVI procedure on the SNP-GPFA model by showing accurate recovery of signal and noise latent structure from simulated SNP-GPFA data. Here, these simulated data consisted of 20 trials of 30 simulated Poisson neurons and were generated from a two dimensional signal and two dimensional noise subspace. Figure 3 shows that our inference procedure accurately recovers the signal (**B**) and noise (**C**) subspace from these simulated data. Only a single trial of the noise subspace is shown for clarity.

To show that our model achieves good performance on real neural data, we fit the SNP-GPFA model to an example recording from the rodent dataset. We found that the the signal component successfully captures the PSTH, or mean response across trials (Figure 4**A**). Importantly, the model also identifies a noise component that accurately predicts trial-by-trial spiking variability. Figure 4**B** shows 3 example trials for the top neuron in **A**. The per-trial rate deviations, given by a $\mathbf{w_{n,i}}^{\top}\mathbf{X_n}$, where $\mathbf{w_{n,i}}$ is an isolated row of $\mathbf{W_n}$, accurately capture per-trial spiking deviations. This can be easily seen in trial three, where a sharp burst at the end of the trial is captured by the noise component of the model.

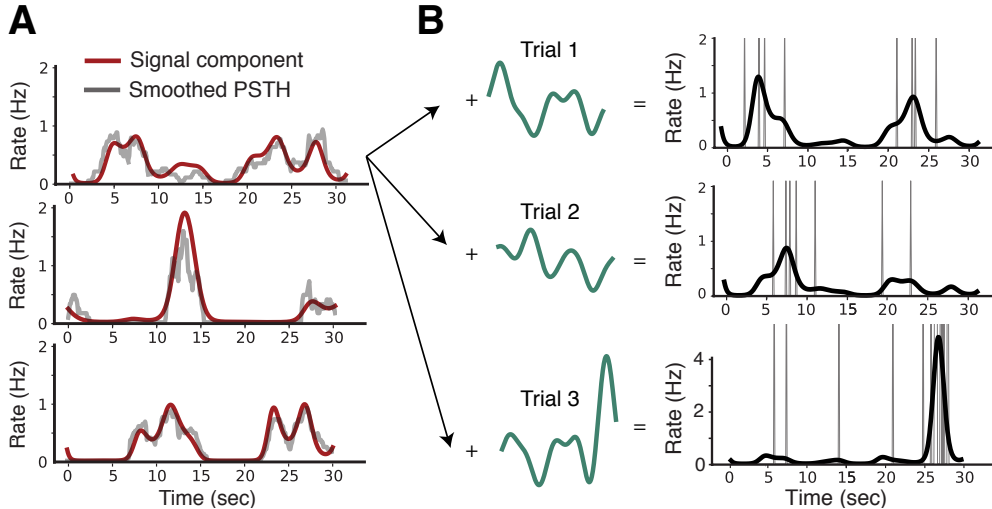

Figure 4: (**A**) Reconstructed PSTHs for three example rodent V1 neurons given by signal model component. (**B**) Trial-by-trial variability explained by noise component for three trials for the top example neuron in **A**.

## 3.1 Learning dimensionality

To identify the dimensionality of signal and noise latents, we used a cross-validation procedure known as co-smoothing [27]. We first train on a subset of randomly selected trials (10 for rodent data, 20 for primate data) and used Fourier-BBVI to learn $\mathbf{W_s}^{\top}$, $\mathbf{W_n}^{\top}$, $\mathbf{X}^s$, and $\ell_s$ ($\ell_s$ here refers to the length scales associated with the signal latents). To test the accuracy of these learned parameters, we withhold a small random selection of neurons and then learn the noise latents $\mathbf{X}^n$ and noise length scales $\ell_n$ over the held-out trials. We evaluate the cross-validated log-likelihood of the held out neurons using these new noise latents and the inferred structure from the initial trials. We perfomed five-fold cross validation and averaged over folds. For additional information about the data preprocessing and cross-validation, see supplemental materials.

We find the signal dimensionality by first leaving out the noise component, and increasing the number of signal dimensions until there is a decrease in CV performance. We then incrementally increase the number of noise dimensions until CV performance decreases. Interestingly, noise components increased CV performance suggesting population-level structure in trial-to-trial variability. For the primate data, this procedure identified 5 signal dimensions and 6 noise dimensions. (See section 3.3 for details of dimensionality for the multi-region rodent data.)

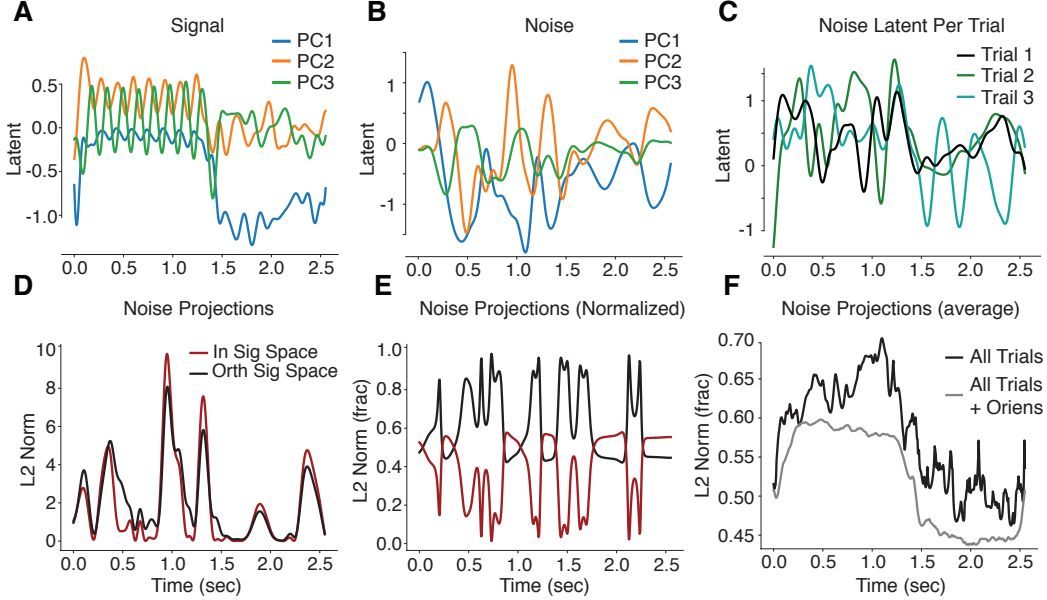

Figure 5: Visualizing noise and signal subspaces. First three PCs of signal (**A**) and noise (**B**) subspaces for 20 trials of a single stimulus presentation. (**C**) The first PC of the noise latents for three random example trials. (**D**) Total L2 norm of noise activity projected into signal subspace (red) and orthogonal to signal subspace (black) for an example trial. (**E**) Same as **D** but normalized to show percentage of noise activity variance. (**F**) Fraction of noise variance orthogonal to signal subspace averaged over all trials for a single orientation (black) and all trials and orientations (grey).

## 4 Results

### 4.1 Visualizing signal and noise subspaces

We use the SNP-GPFA model to visualize the resulting signal and noise subspaces for neural population data recorded from primates. For this experiment, a drifting gratings stimulus was present for the first half of the of a 2.5 second trial (see [25] and the supplemental materials for more details). For a particular stimulus orientation ($0°$), we show the first 3 PCs of the five dimensional signal subspace. Here, we only show three PC dimensions for clarity. Note that during the stimulus presentation, there is a strong sinusoidal component to the latent neural activity. However, after the stimulus presentation period, this structure is no longer present. This latent signal structure in our SNP-GPFA model is nearly identical to the results of P-GPFA run on the PSTH (see Supplemental Figure 1 and [10]). However, unlike P-GPFA, SNP-GPFA additionally extracts a noise subspace (Figure 5**B**). This subspace has no obvious structure and does not include the same sinusoidal component in the first half of the trial. As expected, plotting the first noise latent PC across three example trials, there is no obvious pattern to the noise deviations, reflecting idiosyncratic variations in population firing rates across trials (Figure 5**C**).

An important question that arises with this model is whether or not the noise subspace overlaps with the signal subspace. Overlap of these subspaces implies that trial-to-trial variability in the noise components can corrupt the population response along the signal dimensions, thereby interfering with representation of the signal. Previous work [15] suggests that noise only interferes with the signal reprsentation if it lies in a direction defined by the derivatives of neural tuning curves, and recent work suggests that noise and signal subspaces may indeed be nearly orthogonal [14, 28, 29]. Because our model contains separate latent components for signal and noise, we can explicitly compare the relative angle between these subspaces. More specifically, we look to assess how strongly the pure-noise component of neural activity projects into the signal subspace. Under the SNP-GPFA model, the noise-subspace component of neural activity is $\mathbf{Z_n} = \mathbf{W_n}^\top \mathbf{X_n}$. To assess overlap with the signal subspace, we compute the singular value decomposition of the signal-component loading weights, $\mathbf{W_s} = \mathbf{USV}^\top$, which provides a basis for the signal subspace via the columns of $\mathbf{V}$. The portion of

variance of the noise within the signal subspace for each time point is then given by $||\mathbf{V}\mathbf{V}^\top\mathbf{Z_{n,t}}||_2^2$, where $\mathbf{Z_{n,t}}$ is the $t^{\text{th}}$ column of $\mathbf{Z_n}$, or the L2 norm along the six noise dimensions. The portion of variance orthogonal to the signal subspace is thus simply $||\mathbf{Z_{n,t}} - \mathbf{V}\mathbf{V}^\top\mathbf{Z_{n,t}}||_2^2$.

Figure 5**D** shows the resulting L2-norm time-series both within and orthogonal to the signal subspace. This is for a single example trial for the same single stimulus as in **A-C** (orientation of $0°$). To visualize the fractional variance into and out of the signal subspace, we normalize each trace by the total variance at each time point, $||\mathbf{Z_{n,t}}||_2^2$ (Figure 5**E**). For this trial, the noise activity tends to mostly lie in the subspace orthogonal to the stimulus activity. However, this is not true when the overall noise variance is high. At these moments, the noise exists mostly in the signal subspace. Figure 5**F** shows the fractional noise variance orthogonal to the signal subspace averaged over all trials for the orientation of $0°$ (black line) and all trials and orientations (grey line). It is primarily during the stimulus presentation time that the noise activity is preferentially orthogonal to the stimulus subspace. When there is no stimulus, after the halfway point of the trial, there is a slight preference for the noise activity to lie in the signal subspace.

## 4.2 Shared and independent noise in multi-region data

The rodent dataset we examined contained data from two simultaneously-recorded visual cortical regions, an upstream area "V1" and a downstream area "AL". The SNP-GFPA model can therefore be extended to allow for a characterization of shared variability across these regions. For simplicity, let's consider two versions of the model: (1) a "shared-noise" model, which is the SNP-GPFA model applied to both regions simultaneously (see eq 4); or (2) an independent noise model, which includes a block-diagonalization of $\mathbf{W_n}$ into a V1 component $\mathbf{W_n}^{V1}$ and AL component $\mathbf{W_n}^{AL}$. The independent noise model describing neural activity for trial $j$ is thus:

$$\begin{bmatrix} \mathbf{y}_j^{V1} \\ \mathbf{y}_j^{AL} \end{bmatrix} = \text{Poiss}\left( f(\mathbf{W_s}^\top\mathbf{X}^s + \begin{bmatrix} \mathbf{W_n}^{V1} & 0 \\ 0 & \mathbf{W_n}^{AL} \end{bmatrix} \begin{bmatrix} \mathbf{X}_j^{n,V1} \\ \mathbf{X}_j^{n,AL} \end{bmatrix}) \right) \tag{5}$$

where $\mathbf{X}_j^{n,V1}$ are noise latents that map exclusively to V1 activity, and $\mathbf{X}_j^{n,AL}$ are noise latents that map exclusively to AL activity. By contrast, departures from a block-diagonal structure in the loadings matrix reflects the degree to which latent variability is shared across brain regions.

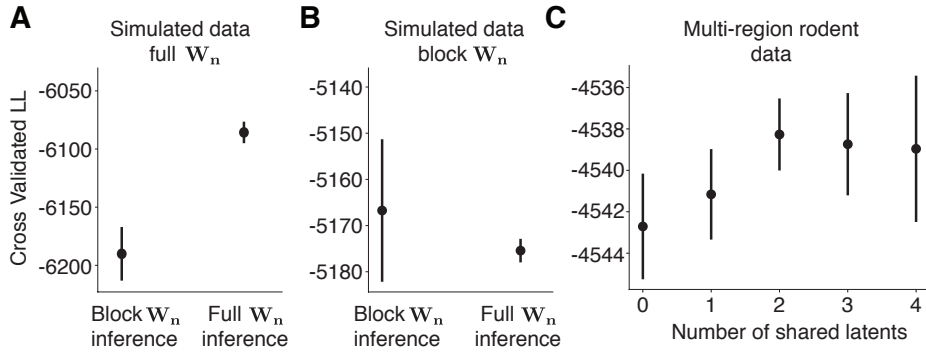

Figure 6: Two-region SNP-GPFA model. (**A** and **B**) Validation of our cross-valdiation (CV) approach on simulated data. (**C**) CV results on real neural data with varied block-diagonal structure for $\mathbf{W_n}$. Bars indicate standard error over 5 fold CV.

We first validate the approach on simulated data. We generate two datasets, one with a full $\mathbf{W_n}$ matrix, and this other with block-diagonal structure. We perform inference on each of these datasets, one using the original model outlined in eq 4, and the other using the block diagonal structure in eq 5. We show that the model with its corresponding generated data exhibits higher CV performance (Figure 6 **A,B**). For additional details see supplement.

On the multi-region rodent data, we compare the SNP-GPFA model with a block-diagonal $\mathbf{W_n}$ to models where $\mathbf{W_n}$ has an increasing number of shared noise latents. We determine a six dimensional signal component. For the noise dimensionality, we start with a complete block representation (eq 5) of 5 V1 dimensions and 4 AL dimensions, which contain no shared components. We then compare

CV performance of this model to ones where increasing numbers of noise latents are shared between the regions (Figure 6**C**). For information regarding how we select the proper number of noise and signal dimensions in this framework, see supplemental materials. We find that models with at least two shared noise dimensions perform better than models with one or fewer shared noise dimensions. Means and standard error shown over five-fold CV (Figure 6). This suggests that there is trial-varying structure in neural population activity that is shared across cortical regions.

## 5  Conclusion

We have introduced a Gaussian process factor analytic model for spike train data that extracts separate signal and noise latent structure from trial-structured data. To learn this model we employ a novel inference method based on black-box variational inference in the Fourier domain, which allows for faster and more stable inference by diagonalizing the posterior covariance and pruning unnecessary frequencies. The resulting SNP-GPFA model is able to extract signal latents that characterize population PSTHs in real neural data, and noise latents that capture trial-to-trial variability that is shared across neurons. We would like to mention that this is not the first model to include a trial varying GP latent with a stimulus fixed component [30, 16]. However, this is the first model we are aware of explicitly designed to uncover separate signal and noise latent subspaces with varying dimensionality. We go on to use the results of the model fit to suggest answers to scientific questions about trial-based neural data. We find that in monkey V1, noise activity tends to project primarily in a subspace orthogonal to signal activity, especially when a gratings stimulus is present, suggesting an optimal type of neural encoding [15]. We additionally use our model on multi-region rodent data and compare performance where noise is shared between cortical regions as opposed to being independent to each region. We find that, for these rodent data, noise models with shared structure better predict held-out spike trains, suggesting variability in the spiking activity is shared across cortical regions. Overall, the model is a promising method for understanding the relationship between stimulus-locked and trial-varying neural activity at the population level. We believe there is a great number of additional scientific questions that the SNP-GPFA model can help answer, including determining how signal and noise representations relate to behavior, and further exploring how block-diagonal loadings matrices may partition signal and noise latent representations in multi-region data. We provide downloadable code for the community to use the SNP-GPFA model on their own trial-based neural data.

## Broader Impact

Here, we propose a new model for neuroscientists to uncover latent structure in trial-based neural population data. Trial-based neural recordings with identical stimuli are ubiquitous in neuroscience research. However, trial-by-trial variability in neural activity is not well understood. More broadly, it is unclear in general what the function of neural noise is in the brain. Our model works on neural population data to separate out neural noise latent representations from stimulus-locked representations. It additionally uses a novel inference technique that is rapid and stable. Here, we provide a general, easy-to-use tool for neuroscientists, and we hope others are encouraged to employ it to understand trial-based neural information in their own experimental set-up. We provide code for download here: `https://github.com/skeeley/SNP_GPFA`. We do not foresee any negative consequences to society resulting from this work.

## Acknowledgements

SLK was supported by NIH grant F32MH115445-03 , MCA and JWP were supported by grants from the Simons Collaboration on the Global Brain (SCGB AWD543027), and a U19 NIH-NINDS BRAIN Initiative Award (5U19NS104648). JWP was also supported by the NIH BRAIN initiative (NS104899 and R01EB026946). YY and SLS were supported by the NIH (R01EY024294 and R01NS091335), the NSF (1450824 and 1707287) the Simons Foundation (SCGB 325407) and the McKnight Foundation.

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
