[Supplementary Material]

# Supplementary Material for "Identifying signal and noise structure in neural population activity with Gaussian process factor models"

## Additional results

### The P-GPFA latent space and the signal subspace of SNP-GPFA

We show here that the signal subspace in the SNP-GPFA model looks nearly identical to standard P-GPFA run on trial-averaged data. This is shown for the same data analyzed in Figure 5 in the main text. The signal latent dimensionality is 5 for the SNP-GPFA model, and the latent dimensionality is 5 for the P-GFPA model. We show the first 3 PCs for clarity. This nearly identical pattern in the subspaces suggests that the SNP-GPFA model is an extension of the P-GPFA model on trial averaged data, providing the same signal subspace as well as additional information about the noise subspace (see Figure 5 in main text for more info).

Figure 1: First 3 PCs of signal subspace in SNP-GPFA and first 3 PCs of the latent space of P-GPFA

## Additional details about simulated and neural data

### Details about primate and rodent data

**Primate Data:** The primate data we use in this manuscript comes from [1]. These are single unit electrophysiological recordings from monkey V1 as they view 72 drifting sinusoidal stimuli. The entire dataset consisted of 113 neurons and 50 trials per stimulus. Here, we only select neurons with greater than 16000 spikes throughout the entire duration of the recording for all stimuli and trials (at least 4.4 spikes per trial on average). This pruned our dataset to 65 neurons total. The reason we select for high firing rate here is because the low-rate neurons are largely uninformative contributors to the noise latent structure due to low trial-by-trial variability, and fewer neurons allowed for faster inference. Additionally, we only use the final 35 trials for our analysis. Because we have to learn an n-dimensional latent structure (n ranging from 1 to 7) per-trial, discarding some trials helped speed up inference of SNP-GPFA.

For cross-validation analysis, we took the 35 trials and randomly divided into 20 train and 15 test trials. Using the co-smoothing procedure described in the manuscript, we learned the noise latents on the held-out trials after withholding two neurons. We then evaluate the log-likelihood on the two held out neuron's 15 held-out trials. We did this over a 5-fold shuffle over held-out trials.

We bin the spikes at 5 ms resolution before performing analysis.

**Rodent Data:** The rodent data comes from a multi-region two-photon imaging set-up described in [2]. Here, we use data from V1 and AL regions in visual cortex in the 'gratings' stimulus condition described in [3]. These stimuli were 20 repeated trials of 8 4-second flashes of orientated drifting gratings presented at 0.05 cycles per degree and 2 Hz. The rodents were head-fixed and passively viewing stimuli that were identical on each trial. Calcium traces from these data were de-convolved to yield spike-times which were subsequently binned at 100 ms resolution.

The full dataset consisted of 352 V1 and 163 Al neurons, but their firing rates were very low; some neurons only spiked 1 or 2 spikes for the duration of the recording. We pruned the dataset to have at least 70 spikes in the 20 presented trials (at least 3.5 spikes per trial) which yielded 30 V1 and 37 Al neurons. All multi-region analysis was done on these 67 total neurons.

Cross-validation procedure was done as described before, here withholding 10 random trials and 2 neurons, one neuron from V1 and one neuron from AL. Means and standard error were again averaged over 5-fold cross-validation.

### Details about multi-region analysis

**Simulated data:** To validate that our Fourier-BBVI is able to distinguish models with block-diagonal $\mathbf{W_n}$ and full $\mathbf{W_n}$ we first generate data from each model. We simulate 20 trials of 24 Poisson neurons. The signal dimensionality is 2 in each condition. However, for the noise dimensionality, in one condition 2 noise latents each map to all of the neurons, and in another, 2 noise latents map to half of the neurons, and a separate 2 noise latents map to the other half. This is used to distinguish two brain "regions". We next perform inference on these simulated data using both models. We withhold 10 trials and two neurons (again, one from one region, one from the other), and calculate log-likelihood on the held-out neurons and trials. We plot means and standard error averaged over five-fold cross validation for each model.

**Selecting signal and noise dimensionality for rodent data:** Identifying the proper signal and noise dimensionality for multi-region data presents a challenge as we cannot check all combinations signal and noise dimensions per-region. To select the dimensionality for this paper, we proceed by first analysing each single region, and then use that information to test the multi-region model. That is, first we identify the optimal signal dimensionality of the V1 data alone and to the optimal signal dimensionality of the AL data alone, determined via averaging over five-fold cross validation. We determine 3 signal dimensions are optimal for each of these conditions. For the multi-region analysis, we select a signal dimensionality that is equal to each of these added together: six dimensions total. For the noise, we similarly independently determine the best noise dimensionality on AL and V1 alone. This was determined to be 5 and 4, respectively. We then run inference on the multi-region data with 6 signal dimensions, and block-diagonalized 5 and 4 noise dimensions.