[Reviews · NeurIPS 2020]

Review 1

Summary and Contributions: The authors propose a few extensions of point process GPFA model and introduce accelerated inference methods by performing computations in the Fourier domain.

Strengths: - Extends a very popular model that extract interpretable summaries of neural dynamics - The method extracts and separates consistent structure from single-trial variability, which is a useful decomposition for exploratory data analysis - Is well-written, provides open source code

Weaknesses: - Other papers have done somewhat similar things, e.g. Duncker & Sahani (2018) have condition-dependent latents as well as single-trial latents. - The paper is somewhat light on experimental results, and the conclusions are somewhat abstract (on *average* there is less overlap between noise and sensory coding dimensions, but there seems to be substantial trial-to-trial variability) - I don't have any other major concerns besides the typical complications and drawbacks of Gaussian Processes. It would be useful to consider different choices for the GP kernel function. Also more discussion on how to choose parameters like the lengthscale of the kernel would be useful. (In the original GPFA paper the authors place a prior over the lengthscale... Is that done here or is the lengthscale a user-determined constant?)

Correctness: Mostly looks good to me. The notation is a bit confusing on page 6 where "Y" is defined to be a linear readout of the latents, but this is different than the "y" on page 5 equation 4 which is the observed spike count.

Clarity: Yes.

Relation to Prior Work: The Fourier domain trick seems to be an adaptation of (https://www.jstatsoft.org/article/view/v019i02) -- I think the authors should add that reference at a minimum. I am not aware if the Fourier domain trick has been applied in the context of BBVI before, though. Additional clarification on how this model differs from Duncker and Sahani (2018) would be useful.

Reproducibility: No

Additional Feedback: - I don't have a strong expectation as to what the lengthscale in the kernel should be for the noise components. Are any of the results sensitive to this choice? - Panel 4D seems to suggest that the amount of "noise" in the signal subspace is almost entirely determined by the norm/amount of noise, not whether the noise is orthogonal to the signal subspace or not. This suggests that panel 4F isn't too informative in and of itself. It would be more informative to show the overall L2 norm of the noise, averaged over trials. If the noise is higher at the beginning of the trials, there could still be interference with sensory coding, do you agree? - It would seem like one might like to set priors on the readout matrices "W" (or regularize them) such that the readout weights are small for the noise dimensions -- this might allow you to have more noise dimensions without overfitting. - In visual cortex, trial-to-trial gain modulation is prevalant and would likely show up in your noise latents... It might be desirable to build this into your non-noise latent dimensions. See Goris et al. (2014) "Partitioning neural variability." ---- RESPONSE TO AUTHOR FEEDBACK ---- I am overall satisfied with this manuscript and am raising my score to a 7 in light of the additional citations and discussion of prior works that will be added. While there are a large number of variants on the basic GPFA model, and there is some overlap between the proposed model and prior work, I still think this will be a moderately useful advance. The impact of the paper would be greatly improved if the authors were to release code and tutorials (ideally with accompanying neural data) alongside their paper.


Review 2

Summary and Contributions: The authors of this paper develop a method for parsing multitrial neural data into signal and noise components using an an extension to Gaussian process factor analysis (SNP-GPFA). They first develop an inference method using a Fourier-transformed latent representation that is fast and flexible. They then create the SNP-GPFA model and use it to address two scientific questions. They show that that noise operates orthogonally to signal activity in primate V1 and that trial-by-trial variability has shared structure across brain regions in mouse visual cortex.

Strengths: This paper introduces a novel and interesting method very clearly, with ample supporting evidence for the veracity and usefulness of their method (with simulations and scientific findings respectively). The use of simulations to demonstrate that their inference method for P-GPFA and their cross-validation approach for SNP-GPFA can recover the "truth" about multitrial data are extremely helpful and a strength of this work. This method for separating noise vs signal on multitrial data is significant - there is a huge amount of neural data that consists of these types of recordings to repeated trials of a stimulus or behavior. The authors outline the types of questions and findings one could use this method for by including some analyses of real neural data: they demonstrate that noise operates orthogonal to signal activity in primate V1 and that trial-by-trial variability has shared structure across brain regions in mouse visual cortex.

Weaknesses: The main weaknesses to me are: 1) the presentation of the scientific findings, 2) some missed opportunities for more rigorous analyses.  I'll detail each below. I found the scientific results (about noise orthogonal to signal, and shared noise across brain regions) extremely interesting but felt that the authors tried to do/claim too much. These findings were a bit awkwardly presented - are they preliminary findings that demonstrate how this method can be used or are they real scientific results? I don't think the authors made this clear. If the former, that needs to made clearer and some of the bolder claims need to be removed. For example, in lines 273, the authors claim that noise tends to be primarily orthogonal with no caveat that this was true in only one dataset of monkey V1. If the latter, there needs to be more discussion of prior work/background/results. Ideally, these results would be shown on more than one dataset - why can't the orthogonality finding also be shown on their mouse data at the very least?. Additionally, in the supplement the authors detail how they remove low firing neurons so there is enough trial-by-trial variability. This seems an important point and a potential weakness of the method if it cannot accurately capture noise when neurons are low-firing (which does not mean these neurons are unimportant and can be removed). This should be addressed in the main text. I also believe there were some missed opportunities for more rigorous analyses. While the inclusion of simulations was appreciated, I think the simulations could have been even more rigorous, with different parameters of the data explored (for example low vs high firing simulated neurons as discussed above). There was also not a robust simulation for SNP-GPFA specifically, except for the cross-validation method. Since this model is their main contributed method, I would advise moving Figure 2 to the supplement and recreating with SNP-GPFA simulations and inference. In line 206, the authors say that the latent signal structure of SNP-GPFA vs P-GPFA is almost identical - they should show this! I would also have liked to see comparisons of their inference method to traditional "computationally cumbersome" methods. ---- UPDATE AFTER AUTHOR REBUTTAL ---- The authors addressed my concerns very satisfactorily in their rebuttal. With their promised changes to the manuscript, I am changing my score to an 8.

Correctness: Yes, the methods and claims are correct.

Clarity: The paper is extremely well written. I think some rearrangement of the structure could benefit the paper though. For example, I found it a bit confusing that the authors only outlined the reason for using Poisson GPFA at the end of section 2 (lines 142-149) - this should have been at least mentioned in the introduction. I also found it a bit odd that the SNP-GPFA model was briefly outlined at the start of section 3 and then the rest of the section was dedicated to results using the model. I would have made Section 2 about both models and Section 3 a "Results" section (or something similar). The authors should also always specify what error bars are showing (ex Figure 5). Also, just a note to the authors, the colors in the figures were occasionally hard to distinguish, at least on my monitor. This was especially true of Figure 1B, but also applies to Figure 2 A/B. I would alter these colors slightly so they are more distinguishable. Minor typos: "Trail" instead of Trial in Figure 4C.

Relation to Prior Work: The authors discussed relevant prior work to their method and made clear when and how they had built upon it (for example, noting that BBVI already exists and they specifically contribute the Fourier-based inference). As noted above, if the scientific findings are presented as more of a demonstration of the use of their method and the types of questions one can ask, I believe the prior work is sufficient. However, as it is, the authors are making slightly overstated claims about their scientific findings with no discussion of relevant work.

Reproducibility: Yes

Additional Feedback:


Review 3

Summary and Contributions: They used GPFA to model the stimulus-dependent trial-to-trial variability. They used the method to explore the validity of the method over some simulated data.

Strengths: Using fourier based variational inference is an interesting approach to learn the GPFA . It can simplify the learning process using the diagonal factorization of the covariance matrix in the fourier domain. There are other works on studying noise correlation and stimulus-dependent trial-to-trial variabilities using different versions of GPFA. The paper comes short in giving proper reference or relation to the existence body of work such as this paper: ‘State dependence of noise correlations in macaque primary visual cortex’. Despite the fact that they seem to use a novel way of solving GPFA’s I think the paper lacks enough novelty and originality more than an implementation of an existing method a bit differently or at least it is not clearly explained or motivated in the paper.

Weaknesses: However, other than computational practicality, the paper should address any particular consequence of parametrizing the inference problem in the fourier domain. There are also other limitations and assumptions which were taken without giving a broader perspective of their consequences and the possiblitiy of going beyond, as an example is to assume a Poisson distribution for the neural activity.

Correctness: The method is well explained and the derivation seems correct.

Clarity: The proposed method and approach is explained clearly. Limitations and assumptions behined using this method and its implications on the correlation structure could be explained better.

Relation to Prior Work: The paper comes short in giving proper reference or relation to the existence body of work such as this paper: ‘State dependence of noise correlations in macaque primary visual cortex’. Despite the fact that they seem to use a novel way of solving GPFA’s I think the paper lacks enough novelty and originality.

Reproducibility: No

Additional Feedback:

[Author Response · NeurIPS 2020]

We thank the reviewers for their constructive feedback, and were grateful for positive appraisals and helpful comments.
However, we feel that **review #4** contained a significant misinterpretation of the existing literature and its relation to our
work. We would thus humbly ask that **R4** consider adjusting their score, if they are convinced by our response below.

**R1:** *Is the length scale a user-determined constant?* No, GP length scales were inferred by maximization of the ELBO.
This is an important point: we typically found longer length scales for the noise GPs than the signal GP (for both mouse
and monkey data), suggesting the trial-to-trial variability has slower timescale than the stimulus-evoked component.

*Citation for Fourier-domain GPs.* Thanks for the reference; now cited. However, we would like to emphasize that (as
the reviewer points out) this is the first time (to our knowledge) that this method has been used for BBVI, and we have
found that the pruned Fourier-domain variational distribution substantially stabilizes and speeds up inference. It also
preserves temporal correlations (off-diagonal elements of the time-domain covariance), allowing for smooth estimates
of the time-series with few BBVI samples.

*Comparison to Duncker & Sahani (DS).* R1 correctly points out DS includes a trial-varying GP component. However,
DS's model was designed to understand time-warping in a GP framework for a reaching task. Each trial included a
time-warping function that allowed the latents to flexibly adapt to different behaviors. Here, our model has distinct
dimensionalities for signal and noise subspaces to understand trial-varying activity in conditions with exact repeated
stimuli. Our model is used to analyze these independently determined signal and noise subspaces. Further, DS does not
adapt methods to understand trial-varying properties of multi-region data. We will clarify these points in the manuscript.

*Light on experimental results.* See R2 comments below and added Fig.

*Noise+sig subspace comparison to overall noise L2.* This is a good point and we will add a comparison to the figure.
R2 is correct that higher L2 norm in the noise latents on average corresponds to anti-alignment with the signal subspace
(though this doesn't always hold). Also, we disagree with R2's conclusion. When the L2 norm is high, the noise is
anti-aligned with signal, and when it is low, it is slightly aligned with signal. This means even if interference is roughly
constant throughout the trial, there is an overall minimization of noise interference with the signal subspace.

*Regularizing W.* We have actually tried this and found no meaningful difference in the learned latent dimensionality.
Smaller W entries traded off with higher-variance latent trajectories.

**R2:** *"authors tried to claim too much ... that noise tends to be primarily orthogonal with no caveat that this was*
*true in only one dataset".* We agree that our claims are too bold. We have updated the wording in the manuscript.
We claim that our model suggests that in monkey V1 data, during stimulus presentation times, the signal and noise
subspaces tend to be more orthogonal than expected by chance, and that in rodent V1 noise structure is shared across
regions. We now make a stronger call for future directions investigating these types of claims using this model.

31
*"Concern about only selecting high rate neurons".* We agree we failed
to clarify this pre-processing step. This choice was primarily motivated
by the fact that many neurons fired <.1Hz, and thus contributed virtually
no trial-varying information (sometimes firing only 3-4 spikes for the
entire experiment.) However, the model fits are fine even using these
uninformative neurons - rates are typically estimated as negligibly small.
The choice simply allowed for rapid inference with similar results.

Fig R1: Inferred signal and noise (1 tr shown)
latents on simulated data.

*"Opportunities for more rigorous analysis and reorg of fig 2 and 4".*
This is very helpful and we agree. We have replaced Fig 2 and now
show performance of the SNP-GPFA model on simulated data. We
moved the former Fig 2 to the supplement. We instead now show BBVI has accurate recovery of signal and noise
latent structure (Fig R1, A and B) and recovery of neural rates (not shown, but fits are similar to P-GPFA fits in Fig 2)
on simulated data. We will also include panels showing recovered signal structure in SNP-GPFA is nearly identical
standard P-GPFA latent structure.

**R4:** The primary concern seems to be that we do not compare to Ecker et al. 2014. Indeed, Ecker et al. is similar
to our model in that includes a trial-varying GP component. However, this trial-varying GP is confined to be 1D,
and there is no stimulus-locked GPFA. Instead, there are trial-averaged tuning-curves which are trivially the same
dimensionality as the neural population and not governed by a GP. Thus, the Ecker et al model cannot identify signal
and noise subspaces and does not allow for any of the analysis that we do in the paper. Additionally, Ecker et al. does
not have Poisson observations, and there is no multi-region analysis. We do find the work relevant as background and
now cite it. However, the lack of reference to this paper seems unreasonable grounds for rejection.

*There are also other limitations ... to assume a Poisson distribution ...* We use Poisson here as we (and others) show it is
a better description of neural activity than standard Gaussian noise (Fig 2D, now in supplement).

We thank the reviewers for identification of other small concerns, typos and figure comments. These are now addressed
in the current manuscript.

[Meta-Review · NeurIPS 2020]

The paper proposes a method for parsing multitrial neural data into signal and noise components using an an extension to Gaussian process factor analysis. The work is interesting, technically sound and has practical applications. A solid paper that is ready for acceptance.